# Long-Term Antibacterial Film Nanocomposite Incorporated with Patchouli Essential Oil Prepared by Supercritical CO_2_ Cyclic Impregnation for Wound Dressing

**DOI:** 10.3390/molecules26165005

**Published:** 2021-08-18

**Authors:** Jingfu Jia, Shulei Duan, Xue Zhou, Lifang Sun, Chengyuan Qin, Ming Li, Fahuan Ge

**Affiliations:** 1School of Pharmaceutical Sciences, Sun Yat-sen University, Guangzhou 510006, China; jiajingfu@mail.sysu.edu.cn (J.J.); zhouxue9@mail.sysu.edu.cn (X.Z.); slf04296923@163.com (L.S.); qinchy8@mail2.sysu.edu.cn (C.Q.); 2School of Traditional Chinese Medicine, Guangdong Pharmaceutical University, Guangzhou 510006, China; 17854223645@163.com

**Keywords:** wound dressing, supercritical CO_2_, patchouli essential oil, antibacterial, long-term, porous, film

## Abstract

Biocompatible skin wound dressing materials with long-term therapeutic windows and anti-infection properties have attracted great attention all over the world. The cooperation between essential oil and non-toxic or bio-based polymers was a promising strategy. However, the inherent volatility and chemical instability of most ingredients in essential oils make the sustained pharmacological activity of essential oil-based biomaterials a challenge. In this study, a kind of film nanocomposite loaded with patchouli essential oil (PEO-FNC) was fabricated. PEO-loaded mesoporous silica nanoparticles (PEO-MSNs) with drug load higher than 40 wt% were firstly prepared using supercritical CO_2_ cyclic impregnation (SCCI), and then combined with the film matrix consisting of polyvinyl alcohol and chitosan. The morphology of PEO-MSNs and PEO-FNC was observed by transmission and scanning electron microscope. The mechanical properties, including hygroscopicity, tensile strength and elongation at break (%), were tested. The release behavior of PEO from the film nanocomposite showed that PEO could keep releasing for more than five days. PEO-FNC exhibited good long-term (>48 h) antibacterial effect on *Staphylococcus aureus* and non-toxicity on mouse fibroblast (L929 cells), making it a promising wound dressing material.

## 1. Introduction

Each year, numerous patients suffer from a variety of skin injuries such as burns, ulcers, fungal infections and other associated traumas, followed with a long recovery period. The development of wound dressings has helped promote wound healing [1,2]. However, the very high incidence of wound infection caused by various microorganisms during the recovery period is still a big problem [3]. Besides the traditional wound dressings, like gauze or cotton wool [4], research on antimicrobial wound dressings in various forms, such as sponges [5,6], hydrogels [7,8], films or membranes [9,10], has gained more and more attention.

An ideal wound dressing requires properties like being non-toxic, good biocompatibility, physical protection, good water absorption and moisture, air permeability, moderate fitting without frequent opening or dressing, stable mechanical properties and producibility of different shapes and sizes [11]. Therefore, natural polymer materials of good biocompatibility and degradability can be effectively applied to the construction of skin repair materials [12,13]. At the same time, further combination of different natural polymer materials can effectively improve the mechanical properties and moisture performance of the dressing [14]. The addition of antibacterial drugs into the dressing materials, such as antibiotics, can improve its antibacterial performance [15,16], but how to maintain the long-term antibacterial function of the dressing, reduce the frequency of dressing replacement and thus reduce the pain of patients, remains an important topic in the treatment of skin diseases. Furthermore, it should be noted that excessive or improper use of antibiotics may easily lead to the emergence of drug-resistant bacteria and increase the difficulty of treatment [17].

Extracts from natural plants or herbs are usually considered of high safety, anti-drug resistance, no side effects or small adverse reactions, and have been applied in various fields for their antibacterial, antiviral and other bioactive effects [18,19]. Among them, essential oils (EOs) are good candidates for active components added in wound dressing materials, due to their high lipophilicity, short molecular chain and thus their good permeability through skin [20,21]. EOs also commonly have unique volatility and aromatic odors that may help relieve the patients’ mood. Patchouli essential oil (PEO) is extracted from *Pogostemon cablin* (Blanco) *Benth* (Lamiaceae), and has a variety of pharmacological activities, including antibacterial, antiviral, anti-inflammatory and anti-tumor effects, as well as enhancing memory and improving gastrointestinal function. Patchouli alcohol and patchouli ketone are considered as the main antibacterial components. However, also because of their volatile nature, EOs usually evaporate too fast to maintain long-term antibacterial effect. Embedding EO into mesoporous nanocarriers is a potential way to control its release behavior in a slow and sustained pattern [22,23], but EO loading into nanocarriers efficiently is still a primary difficulty for this strategy. Based on the strong permeability of supercritical CO_2_, as well as its well solvation property to EO or other active components, the supercritical impregnation process has been used for different applications, such as active food packaging, drug delivery or functional materials [24,25,26,27]. To further increase the drug load of the nanoparticles, our previous work has developed a modified supercritical impregnation method, named supercritical CO_2_ cyclic impregnation (SCCI), to load EOs, including zedoary essential oil [28] and cinnamon essential oil [29], into silica mesoporous nanoparticles with drug loads much higher than those using the normal supercritical impregnation method.

Therefore, in this study, we choose PEO as the antibacterial component and attempt to fabricate a film nanocomposite as a potential wound dressing material, which consists of PEO-loaded nanoparticles and biodegradable film made of poly(vinyl alcohol) (PVA) and chitosan (CS). To our best knowledge, there have been no reports of wound dressing materials containing patchouli oil as an antibacterial substance. The main purposes of this study are to fabricate a long-term antibacterial wound dressing material with good mechanical and moisturizing properties in order to conform to the general requirements for skin dressing products.

## 2. Results

### 2.1. Structure and Morphology of PEO-MSNs and PEO-FNC

The pore structures of MSNs before and after loading PEO were investigated using BET and BJH measurements. Seen in Figure 1A,B are the nitrogen-adsorption-desorption isotherms of MSNs and PEO-MSNs. The curves refer to MSNs (Figure 1A) and the demonstrated distribution of uniform mesoporous channels inside MSNs. In addition, for the curves of PEO-MSNs (Figure 1B), the adsorption/desorption volume reduced significantly compared to that of MSNs, indicating that PEO was successfully loaded into the MSNs. Further information from the BET and BJH tests are listed in Table 1, including specific surface area, pore volume and pore size. MSNs had a high BET specific surface area value of 610.67 m^2^/g, and the BJH adsorption and desorption values including surface area, pore volume and pore width were approximate, demonstrating the well-structured inner mesoporous space. Such large pore volume endowed the MSNs with outstanding drug loading capacity. It was quite different for PEO-MSNs that the BET surface area dropped sharply to 250.18 m^2^/g, attributing to the loaded PEO. The differences between the adsorption and desorption values for PEO-MSNs may be because the loaded PEO volatilized with the nitrogen gas together during the desorption process, thus altered the determined nitrogen partial pressure. The successful loading of PEO in MSNs can avoid its rapid volatilization loss, and provide the structural basis for realizing the long-term antibacterial effect of PEO.

The surface morphology of PEO-MSNs was observed by both SEM and TEM (Figure 1C,D). The images showed that PEO-MSNs were uniform spherical particles with diameters around 50 nm, and the rough appearance on the particle surface indicated the pore channels of MSNs. SEM images of the as-prepared PEO-FNC, both cross section and surface morphologies, are shown in Figure 1E,F. From the cross-section image (Figure 1E), the presence of plentiful porous channels of different micron sizes could be seen, which provided sufficient space for the PEO-MSNs nanoparticles to distribute evenly inside the film. This pore structures also provided channels for the release of PEO from the film nanocomposite. The surface morphology of PEO-FNC in Figure 1F showed that PEO-MSNs nanoparticles dispersed well in the film. This homogeneous distribution was crucial for the uniformity of the film quality, such as its mechanical properties and the local release behavior of the loaded PEO.

### 2.2. Drug Load of PEO-MSNs and the Effects of SCCI Parameters

The composition of PEO was analyzed by using both HPLC (Figure 2A) and GC-MS (Figure 2B), and the results showed that the major ingredients of PEO were alcohols, alkenes, ketones and some esters. Since most of the volatile ingredients in PEO had no UV absorption during the HPLC analysis, the total amount of PEO was calculated based on the quantitative determination of patchoulenone, whose content was 2.53% in the unprocessed PEO.

The GC-MS fingerprint of PEO-loaded in PEO-MSNs was consistent with that of the unprocessed PEO, showing that the composition of PEO remained stable after the procedure of SCCI. Specifically, several major ingredients with their retention time (unprocessed PEO/PEO-loaded) were marked out in Figure 2B,C, such as patchouli alcohol (52.14/52.17 min), phthalene (32.03/32.07 min), α-guaiene (36.39/36.44 min), γ-patchoulene (38.41/38.47 min), and δ-guaiene (42.69/42.75 min). Their peak area proportions of the total peak areas (unprocessed PEO/loaded PEO) were 43.78%/43.35%, 16.03%/15.68%, 4.69%/4.70%, 7.59%/7.61% and 11.86%/11.96%, separately. The values between unprocessed PEO and loaded PEO had no significant difference according to T-test. Therefore, the drug load of PEO-MSNs was calculated by determining the amount of patchoulenone in the following experiments.

The effects of SCCI operating parameters including pressure, loading time and cycle index (the running number of cycles) on drug load of PEO-MSNs were investigated, and the experiments carried out with the drug load of products can be found in Appendix A. To understand more visually, Figure 3 shows the change laws of drug load according to the operating parameters. When the loading time was fixed at 60 min with only one cycle, the drug load increased from 19.9 ± 0.3% at 12 MPa to 28.5 ± 0.3% at 15 MPa, and flattened at a top level with further pressure increase (Figure 3A). This might be because the diffusive equilibrium had not been achieved within the loading time set at lower pressure (12 MPa). This conclusion was confirmed by the results of drug load change according to loading time (Figure 3B). At 12 MPa, the drug load further increased to 27.8 ± 0.6% that was close to the top level when the loading time extended to 90 min. Besides, when the loading time was decreased to 30 min, the drug load was decreased to 24.3 ± 0.7% even at 15 MPa. It could also be concluded that the dissolving rate of PEO in supercritical CO_2_ or the diffusing speed of PEO/CO_2_ into MSNs was accelerated by pressure rising. In Figure 3C, it could be found that the drug load was further improved by adding cycle index, and the drug load reached a top level after enough cycles. This tendency was consistent with the results of preparing MSNs nanoparticles loaded with zedoary oil, which had been reported in our previous study [24]. Generally, after PEO was loaded in the pore channels inside MSNs via a cycle, because of the strong hydrogen bond interactions caused by the silanol groups and the capillary action of mesoporous channels, partial PEO would hardly dissolve in supercritical CO_2_, or its dissolution rate would decline sharply in the next circle. Thus, more PEO would be able to diffuse into the channels, resulting in higher drug load. Here, when the cycle index was 4, with pressure of 15 MPa and loading time of 60 min, the drug load reached an almost saturated value of 42.8 ± 0.5%. Thus, the conditions of pressure of 15 MPa, loading time of 90 min, and cycle index of 4 were used for the experiments that followed.

### 2.3. Physical and Mechanical Properties

As the hygroscopicity is a very important property for an ideal skin dressing material, the hygroscopic weight increments of PEO-FNC with different PEO-MSNs contents were investigated first (Figure 4A). The results showed that all PEO-MSNs products had good hygroscopicity. When put into PBS solution, the film nanocomposites could absorb water rapidly to a saturation level within 10 min, and their weights increased more than 200%. Besides, it was observed that the hygroscopicity was weakened by the increase of nanoparticle contents. This may be because the nanoparticles adhered on the inner surface of the pore channel inside the PVA/CS film and hampered part of its hydroxyl groups to form hydrogen interactions with water molecules. The results demonstrated that PEO-FNC had good swelling capacity and water retention properties to maintain the moist condition required for wound healing.

The thickness of wetted PEO-FNC was between 300~350 μm, and the impact of nanoparticle content on film thickness was inapparent within the scope of this study. TS and EAB% are also important evaluation indicators for skin dressing materials, as a certain mechanical strength is necessary for wound protection and long-term treatment. Figure 4B provides the changes of TS and EAB% according to the nanoparticle content. The TS value of PEO-FNC increased at first with the nanoparticle content and became less steep when the nanoparticle content was higher than 0.10%, while the EAB% kept going up. The results indicated that the addition of nanoparticles improved the film mechanical strength. This might be because the internal hydrogen bonds were formed among the silicon hydroxy, PVA and CS long-chains, enabling the nanoparticles to share the drawing force. A similar trend has been reported in other literatures [25,30]. The addition of PEO had little influence on the mechanical properties of the film due to its relative low amount and inner entrapment inside the MSNs.

### 2.4. Release Behavior of PEO from PEO-FNC

The release behavior of PEO has a direct bearing on the therapeutic window of the PEO-FNC as a skin dressing. Figure 5A showed the change of accumulative released amount of PEO in PBS (pH = 7.4) solution according to the time. PEO-FNC exhibited well slow-release effect on PEO and kept for more than five days. After a 5-day release, the accumulative released amounts of PEO for nanoparticle contents of 0.05%, 0.10%, 0.20% and 0.30% (these percentages refer to the initial mass ratios added during the preparation of PEO-MSNs) were 82.7%, 85.0%, 61.6% and 46.5%, respectively. The percentages were calculated based on the drug load of PEO-MSNs and absolute contents of PEO-MSNs in final produced films. The mass of final film was about 7.0% of the initial gel-like mixtures, and the absolute PEO-MSNs contents in final film was 0.71%, 1.43%, 2.86% and 4.29% in sequence. Figure 5B compared the stage released amounts of different nanoparticle contents within each time interval. It could be seen that the stage released amount increased with nanoparticle content when the latter was lower than 0.2%, but flattened at higher nanoparticle content. This might be because, although the content of nanoparticles loaded with PEO increased, the maximum flow rate inside the pore channels of the film nanocomposite was limited, resulting in a saturated release amount in unit time. Based on the data from the PEO release behavior study, the maximum release rate of PEO from PEO-FNC could be estimated as 62.5 μg·h^−1^·g^−1^.

Besides, when the PEO was added directly into the PBS solution, the liquid had obvious stratification, and it was difficult to detect the composition of PEO in aqueous phase, indicating that PEO was insoluble in water. In the process of determination of the release curve of PEO-FNC, the content of essential oil entered the aqueous phase and could be effectively determined, which indicated that the dissolution rate of PEO was significantly enhanced.

### 2.5. Cytotoxicity Exhibition of PEO-FNC

From Table 2, the concentration at 75.3% cell viability of patchouli oil was 12.5 µg/mL, which was far higher than the amount of oil in 0.3% PEO-FNC. The L929 cells in contact with 0.30% PEO-FNC exhibited nontoxicity and good biocompatibility. The cell viability of MSNs was 84.7% at the concentration of 20.0 µg/mL, which was much higher than the contents in the film nanocomposite. The results indicated that the PEO-FNC prepared was safe and nontoxic.

### 2.6. Antibacterial Properties

The antibacterial effects within 48 h of the PEO-FNC as well as the unprocessed PEO and blank film were compared, and the results are shown in Figure 6. For *S. aureus* in culture solution without any sample added (the control group), the logarithmic phase started early before 4 h and last until 20 h. After 24 h of growth, the order of magnitudes of the *S. aureus* colony amount reached 14 and remained stable for the following 24 h. For *S. aureus* treated with blank film (no PEO-MSNs in the film), although the logarithmic phase was postponed nearly 4 h, which might be caused by the slight antibacterial effect of chitosan and PVA, the bacteria began to grow fast in the following time and the colony amount reached the same level as the control group finally. When adding PEO directly into the bacterial culture medium, the colony amount of *S. aureus* was reduced slightly during the first 8 h due to the antibacterial effect of PEO. However, the logarithmic growth phase appeared at 16 h and the bacteria started to proliferate quickly. There were two possible reasons for this phenomenon: One was that only a small part of PEO added was dissolved in the culture medium and acted on the bacteria due to the poor dissolution property of unprocessed PEO; the other was that PEO was really volatile and most of it was lost rapidly at 37 °C. The curve for PEO-FNC was quite different. It could be seen that the growth of *S. aureus* was limited completely within 48 h and the bacteria amount was decreased gradually at a slow rate. This long-term inhibition effect to *S. aureus* was from the controlled and sustained release of PEO from PEO-FNC.

## 3. Materials and Methods

### 3.1. Materials

Patchouli essential oil (PEO) of pharmaceutical grade with quality inspection meeting the Chinese Pharmacopoeia standard was supplied by Jiangxi Anbang Pharmacy Co., Ltd. (Jiangxi, China). Chitosan (CS, viscosity > 400 mPa·s) was obtained from Macklin (Shanghai Macklin Biochemical Co. Ltd.). Poly(vinyl alcohol) (PVA, alcoholysis degree: 99.0–99.4 mol%, viscosity: 12.0–16.0 mPa·s), tetraethyl orthosilicate (TEOS, >99%), cetyltrimethylammonium bromide (CTAB, >99%), diethanol amine (DEA, >99%), glycerol (>99.7%, GC) and standard substance of patchoulenone (>98%) were purchased from Aladdin (Shanghai, China). Carbon dioxide (99.99%) was obtained from Guangzhou Gas Factory Co., Ltd. (Guangzhou, China). Deionized water was produced by a Milli-Q system and acetonitrile was chromatographically pure. Other reagents were analytical grade and used directly.

Tryptone agar and yeast extract were from Coolaber Science and Technology Co. Ltd. (Beijing, China). Stock culture of *Staphylococcus aureus* (ATCC 27217) was obtained from Solarbio Science and Technology Co., Ltd. (Beijing, China). Minimum essential medium (MEM), fetal bovine serum (FBS), 0.25% trypsin EDTA, Dulbecco’s phosphate-buffered saline (D-PBS, pH = 7.2) and phosphate buffer saline (PBS, pH = 7.4) were purchased from Gibco, ThermoFisher CN (Shanghai, China). Penicillin-streptomycin solution (P/S, 10,000 units/mL penicillin and 10,000 μg/mL streptomycin) was purchased from HyClone, ThermoFisher CN. MTT was purchased from Aladdin (Shanghai, China). L929 fibroblast cell lines were purchased from Procell Life Science and Technology Co., Ltd. (Wuhan, China).

### 3.2. Loading PEO into MSNs via Supercritical CO_2_ Cyclic Impregnation (SCCI)

The MSNs were prepared using the alkaline-medium synthetic method described previously (Appendix A) [31]. Then PEO was loaded into MSNs to obtain PEO-MSNs nanoparticles via a new developed method named SCCI, and the equipment was illustrated in our previous report [28]. In summary, 200 mg of MSNs were firstly placed in a stainless basket with an air-permeable bottom and sealed in a high-pressure drug loading kettle. Afterwards, the following drug loading procedure could be divided into several identical cycles, and each cycle included 3 steps as follow: (1) PEO injection, where 2 mL PEO was injected into the kettle through a nozzle at the top and immersed the MSNs for 15 min. (2) Supercritical impregnation stage, where CO_2_ was delivered continuously until the pressure reached 18 MPa and the kettle was warmed to 40 °C synchronously. In the next hour, PEO was thoroughly dissolved in the CO_2_ under this supercritical state, and a part of it was carried into the pores of MSNs by CO_2_ at equilibrium. (3) Depressurization state, where the temperature was firstly decreased below 20 °C by a water condensing system, and then the CO_2_ and unloaded PEO were released out of the kettle slowly. PEO was recovered for cycle use. After a few cycles, the final PEO-MSNs nanoparticles were collected and stored at 4 °C for the later test. In this study, the running number of such cycle was named cycle index.

### 3.3. Film Nanocomposite Fabrication

Film nanocomposite based on the mixture of PVA and chitosan was prepared by a casting method. Firstly, chitosan was dissolved in 1% (*v*/*v*) acetic acid solution at 60 °C to get a 2% (*w*/*v*) chitosan solution, while 10% (*w*/*v*) PVA aqueous solution was obtained at 92 °C. Then the chitosan and PVA solutions were mixed with a solution weight ratio of 7:3. Afterwards, different amounts of PEO-MSNs (0.05%, 0.10%, 0.20%, and 0.30%, *w*/*w*) were added in to get a series of gel-like suspensions with magnetic stirring of 600 rpm for 10 min. Finally, the suspensions were casted in Teflon molds (100 × 100 mm) and freeze-dried to form PEO film nanocomposites (PEO-FNC). Blank film nanocomposites were also prepared using MSNs without any PEO-loaded.

### 3.4. Characterization

#### 3.4.1. Structure and Morphology of MSNs and PEO-FNC

The morphologies of MSNs, as well as PEO-FNC samples, were observed using scanning electron microscope (SEM, Gemini500, Zeiss/Bruker, Karlsruhe, Germany). Brunner–Emmet–Teller (BET) measurement by a specific surface meter (JW-BK200C, JWGB Sci. and Tech. Co., Ltd., Beijing, China) was used to determine the internal structure of the particles.

#### 3.4.2. Mechanical Properties of Film Nanocomposite

The film thicknesses of PEO-FNC and blank film nanocomposites were determined by a professional digital display thickness gauge (EXPLOIT, Jinhua, China), while their tensile strength (TS) and elongation at break (EAB%) were determined by a fastener tension tester (QJ212, Qingji, Shanghai, China). The samples were wetted under 50% humidity for 48 h and then cut into rectangular strips (5 × 25 mm) before test. All tests were paralleled 10 times.

### 3.5. Contents and Release Behavior of PEO

PEO was determined by both a high-performance liquid chromatography system (HPLC, UltiMate 300, Thermo Fisher Scientific Ltd., Waltham, MA, USA) and a gas chromatograph-mass spectrometer (GC-MS, TRACE 1300, Thermo Fisher Scientific Ltd.). The amount of PEO was calculated based on the determination of patchoulenone by HPLC, and the ingredients distribution of PEO was analyzed through the finger-print map obtained by GC-MS. The detailed description of HPLC and GC-MS can be found in Appendix A.

To determine the drug load of PEO-MSN, the nanoparticles were firstly immersed in methanol with ultrasonic treatment for 2 h, and the resulted solution was determined by HPLC. Then the release behaviors of PEO from PEO-FNC with different PEO-MSN contents in neutral PBS were investigated under room temperature. Concisely, about 1 g PEO-FNC was precisely weighted and put into 50 mL PBS (pH 7.4) with 150 rpm magnetic stirring. At each preset time interval, 1 mL solution was withdrawn from the same water level for HPLC determination, and 1 mL PBS was replenished immediately. The release assays were done in triplicate.

### 3.6. Cytotoxicity Test

Cytotoxicity of PEO was determined via MTT (3-(4,5)-dimethylthiahiazo (-z-y1)-3,5-di-phenytetrazoliumromide) assay. Mouse fibroblast (L929 cells) was cultured in minimal essential medium (MEM) supplemented with 10% fetal bovine serum (FBS), 1% P/S at 37 °C and 5% CO_2_. The 0.30% PEO-FNC and blank film (film without PEO-MSN) were cut into 2 × 2 cm and wiped with 75% ethanol before sterilized by 1 h UV radiation. Subsequently, the films were fitted at the bottom of each well in the 6-well plates under strict aseptic conditions. MEM medium (5 mL) containing cell suspension was added to 6-well plates, at a density of 2 × 10^4^ cells/mL. The morphology of cells was observed under microscope. L929 cells were plated in 100 µL of MEM, which was supplemented with 10% FBS and 1% P/S, at a density of 5000 cells/well in a 96-well plate. After 24 h, culture medium was removed and varying concentrations of PEO (1.5625, 3.125, 6.25, 12.5, 25, 50 µg/mL) and MSN (5, 10, 20 µg/mL) were added. The cells were incubated for 48 h, 20 µL of 5 mg/mL MTT was added for 4 h at 37 °C and 5% CO_2_, and the absorbance was determined at 490 nm. L929 cells without any treatment were used as the negative control.

### 3.7. Antibacterial Activity

The antibacterial activities of PEO-FNC (0.30% PEO-MSNs added), as well as blank film and PEO alone, were investigated against *Staphylococcus aureus* (*S. aureus*, LA9190, Solarbio, Beijing, China) using a viable colony count method [32]. The solid medium was made by mixing 1000 mL distilled water, 10 g tryptone, 5 g yeast extract, 10 g sodium chloride and 15 g AGAR, and pH was adjusted to 7–7.4, then heated and boiled until solidified. The resulted medium was divided and sterilized for later use. The liquid medium was a mixture of 1000 mL distilled water, 10 g tryptone, 5 g yeast extract and 10 g sodium chloride. After heating and boiling, the pH was adjusted to 7–7.4, and the medium was divided and sterilized for later use.

The antibacterial method could be described as follow: Firstly, the bacterial solution was diluted to 1.5 × 10^5^ cfu/mL, and 50 μL solution was taken and inoculated in 5 mL liquid medium. After that, a 60 mg film sample or 1.50 mg PEO (a little more than the PEO amount of 1.12 mg in PEO-FNC sample used) was put into the liquid medium and cultured at 37 °C in an incubator. Then, 100 μL samples were withdrawn every 4 h and diluted by appropriate amounts, and then coated on solid medium to determine the number of bacterial colonies. Bacterial solution without anything added was used as control.

## 4. Conclusions

A film nanocomposite with PEO-loaded was fabricated, using supercritical CO_2_ cyclic impregnation (SCCI), as a kind of skin dressing material with long-term antibacterial effect and non-toxicity. The film nanocomposite consisted of PEO-loaded silica nanoparticles and a PVA/CS film matrix. The drug load of PEO-MSNs was up to 42.8 ± 0.5%, benefiting from the high loading efficacy of SCCI. The SEM and TEM observations showed that PEO-MSNs around 50 nm dispersed homogeneously in the micro porous matrix of the PVA/CS film. The mechanical characterization for PEO-FNC demonstrated that the as-prepared film nanocomposite had a good hygroscopicity and the weight increase rate was more than 200%. Besides, the TS and EAB% were improved by adding PEO-MSNs compared to blank film. In the drug release test, PEO-FNC exhibited good slow-release effect on PEO and kept for more than five days. Due to this controlled and sustained release property, the PEO-FNC showed a long-term inhibition effect to *S. aureus*, which makes it a promising skin dressing material.

## Figures and Tables

**Figure 1 molecules-26-05005-f001:**
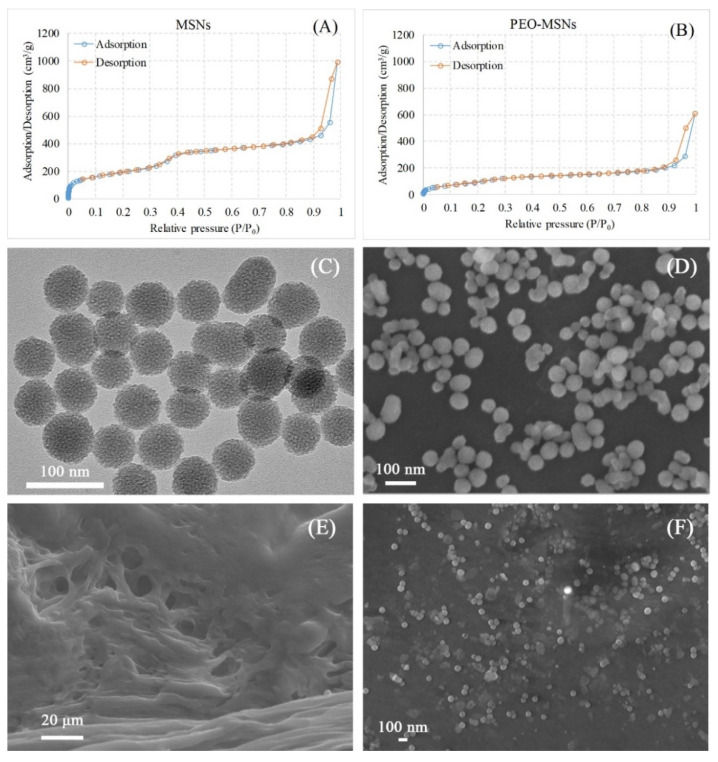
Structures and morphologies of MSNs, PEO-MSNs and PEO-FNC: Nitrogen-adsorption-desorption isotherms of (**A**) MSNs and (**B**) PEO-MSNs, surface morphology of PEO-MSNs observed by TEM (**C**) and SEM (**D**), SEM images of cross-section (**E**) and surface (**F**) of PEO-FNC.

**Figure 2 molecules-26-05005-f002:**
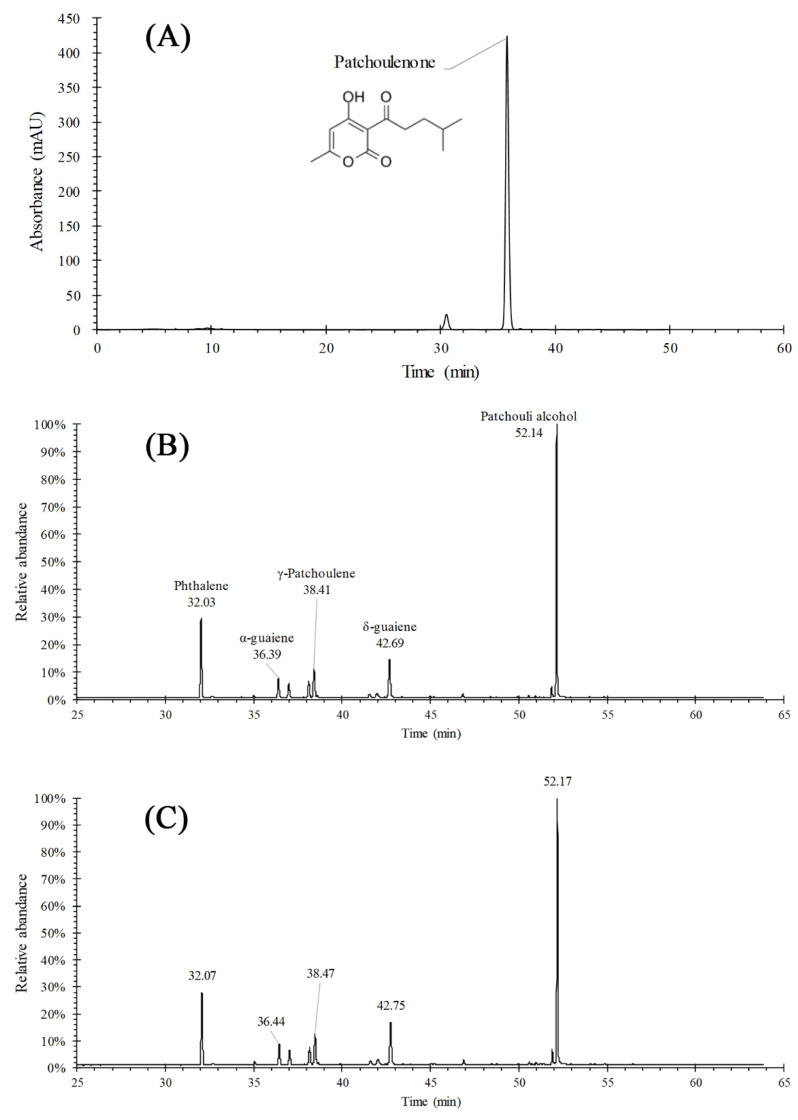
Chromatography detection of PEO: (**A**) HPLC profile and (**B**) GC-MSN profile of unprocessed PEO, and (**C**) GC-MS profile of PEO in PEO-MSNs.

**Figure 3 molecules-26-05005-f003:**
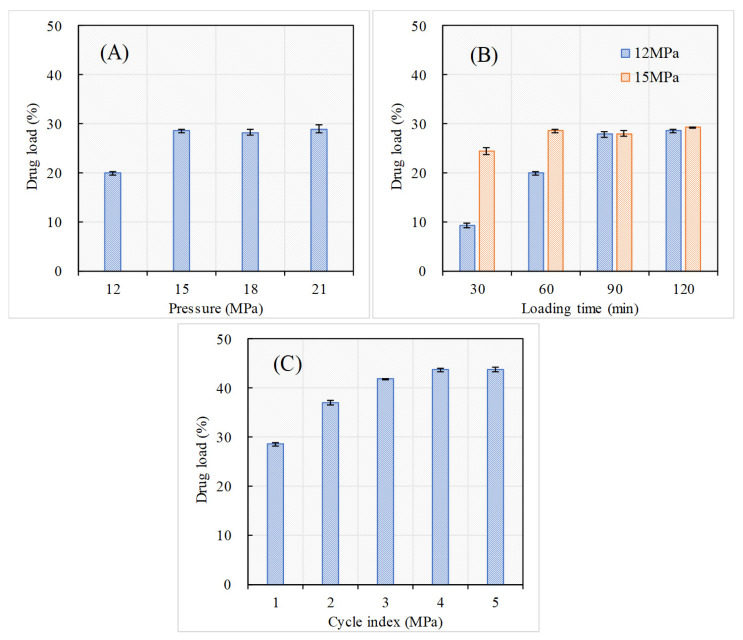
Effects of SCCI operating parameters on drug loads: (**A**) Pressure effect at loading time of 60 min and cycle index of 1; (**B**) loading time effect at pressure of 12 MPa (**left**) and 15 MPa (**right**) and cycle index of 1; (**C**) cycle index effect at pressure of 15 MPa and loading time of 60 min.

**Figure 4 molecules-26-05005-f004:**
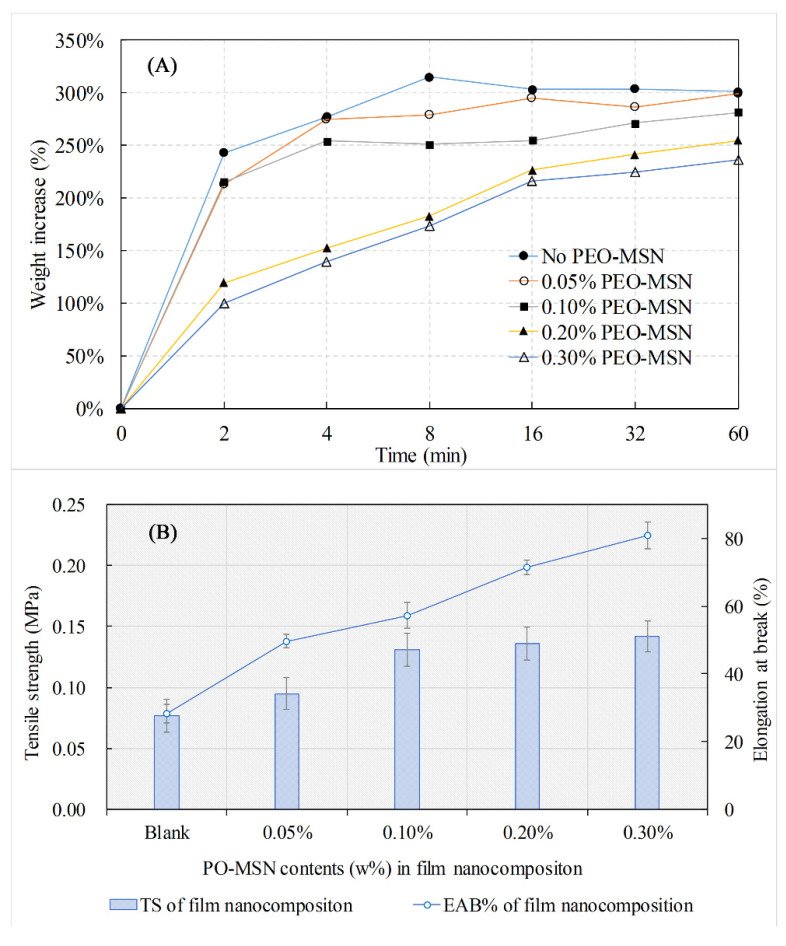
Physical properties of PEO-FNC with different nanoparticle contents: Hygroscopicity (**A**), tensile strength (TS, bars in (**B**)) and elongation at break (EAB%, line in (**B**)).

**Figure 5 molecules-26-05005-f005:**
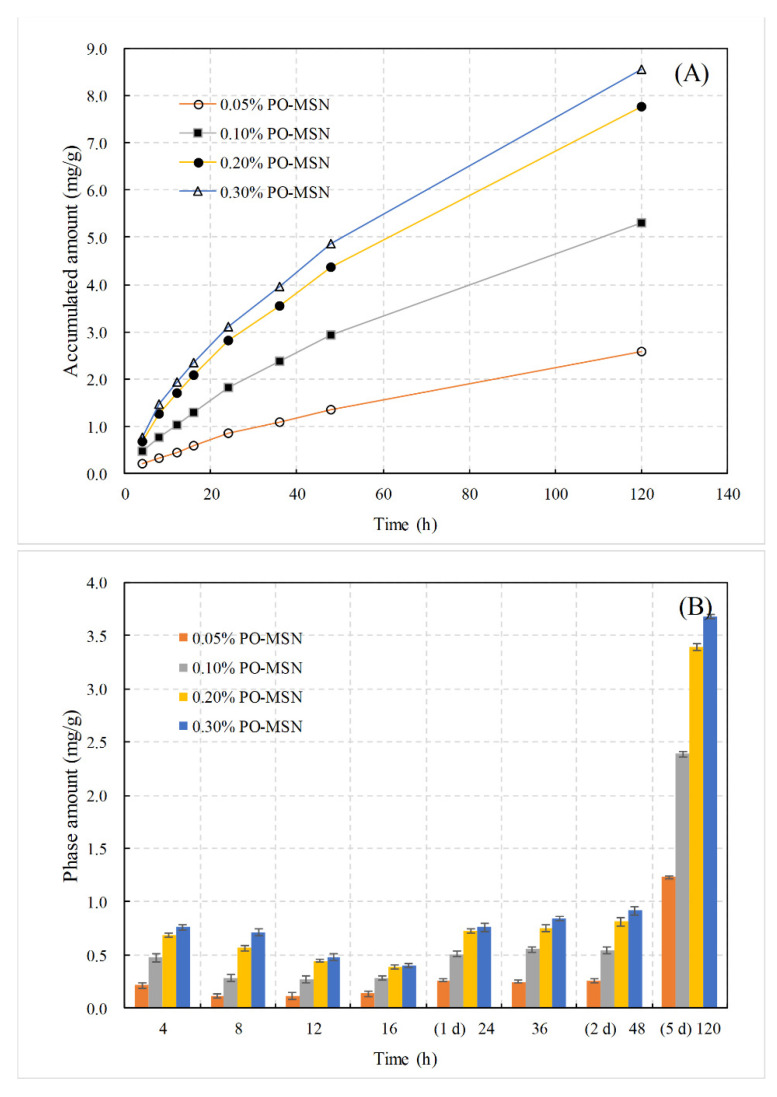
Release behavior of PEO (mg) from 1 g PEO-FNC with different nanoparticle contents in PBS (pH = 7.4) solution. (**A**) Accumulative release amount according to time, (**B**) phase release amount within each time intervals.

**Figure 6 molecules-26-05005-f006:**
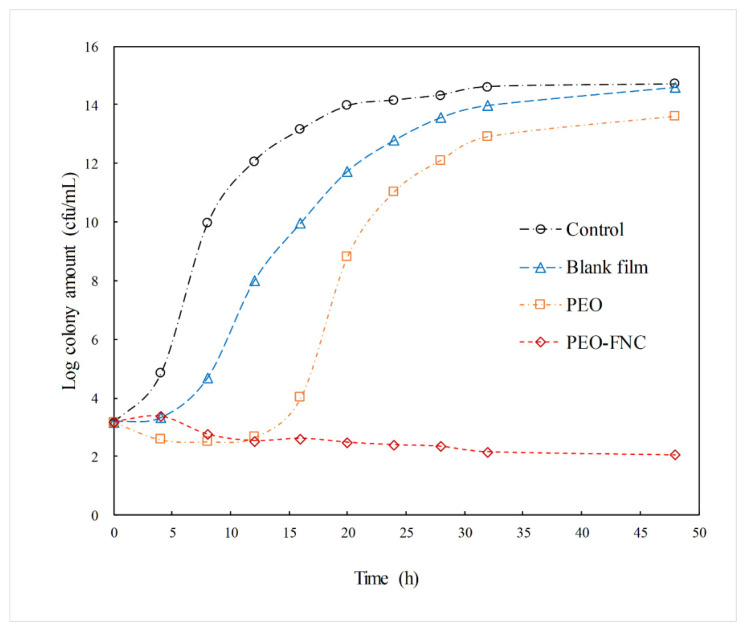
Growth of *S. aureus* in culture solution containing different samples.

**Table 1 molecules-26-05005-t001:** Results of BET and BJH determination for MSNs and PEO-MSNs.

Name	MSN	PEO-MSN
BET surface area	610.67 m^2^/g	259.18 m^2^/g
BJH adsorption surface area	868.68 m^2^/g	396.72 m^2^/g
BJH desorption surface area	873.01 m^2^/g	448.66 m^2^/g
BJH adsorption pore volume	1.58 cm^3^/g	0.45 cm^3^/g
BJH desorption pore volume	1.61 cm^3^/g	0.80 cm^3^/g
BJH adsorption pore width	7.37 nm	4.47 nm
BJH desorption pore width	7.29 nm	7.04 nm

**Table 2 molecules-26-05005-t002:** Inhibition rates of PEO to L929 cells.

Concentration (μg/mL)	1.6	3.1	6.3	12.5	25.0	50.0
Inhibition rate	0.4%	−2.4%	2.6%	24.7%	49.6%	81.3%

## Data Availability

The data presented in this study are available on request from the corresponding author.

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
