# Peer review of "Long-Term Antibacterial Film Nanocomposite Incorporated with Patchouli Essential Oil Prepared by Supercritical CO2 Cyclic Impregnation for Wound Dressing"

_molecules, 2021, doi:10.3390/molecules26165005_

Round 1

Reviewer 1 Report

This article studies the supercritical impregnation of PEO in mesoporous silica nanoparticles for the development of PVA/chitosan nanocomposites for wound dressing. The supercritical impregnation process has been widely used for different applications as for active food packaging, drug delivery and others. The authors should mention in the manuscript these different applications. For active food packaging I recommend to the authors these works:

  1. a) Effect of pressure and time on scCO2-assisted incorporation of thymol into LDPE-based nanocomposites for active food packaging

https://doi.org/10.1016/j.jcou.2018.05.031

  1. b) Modifying an Active Compound’s Release Kinetic Using a Supercritical Impregnation Process to Incorporate an Active Agent into PLA Electrospun Mats

https://doi.org/10.3390/polym10050479

On another hand, the supercritical impregnation process has been carried out by batch procedures, being the most efficient alternative for the incorporation of active compounds into different matrices. In this sense, the authors should mention in the manuscript why they used a cyclic impregnation process and include in the introduction the different alternatives used for the supercritical impregnation (Batch, continuous, etc ….) and comment their advantages and disadvantages. I recommend to the authors read and cite to the following work which explain in detail the different processes used.

Moreover, I have the following suggestion to improve the quality of the manuscript:

  1. - change “of” by “between” in line 13.
  2. - Use “natural” instead “nature” in line 52
  3. - “Depressurization step” instead “depressure state”
  4. – “With magnetic stirring of 600 rpm” instead “with a magnetic stir of 600 rpm”
  5. - release assays were done in triplicate or at least in duplicate?
  6. - Define MTT assay
  7. – The authors should cite the source of the protocol used for the determination of the antibacterial activity.
  8. – Why do you use cycles for the supercritical impregnation instead a common use batch procedure putting an excess of active compound inside the impregnation cell in order to guarantee the maximum gradient concentration for the active compound between the scCO2 and the nanoparticles ?
  9. - The improvement in the mechanical properties of polymeric films by the addition of nanoparticles it is well-known. The authors should compare their results with the reported in other studies. Moreover, the authors should comment about the effect of the essential oil loaded into the nanoparticles over the mechanical properties of the films.
  10. - Change “drugload” by “drug load” in the entire manuscript.
  11. – The sentence “This pore structures were also benefit for the release of PEO from the film nanocomposite” between lines 108-109 should be explained. Why a porous structure in the nanocomposite film should delay the release of the essential oil?
  12. – Which was the CO2 pressure used for obtaining the PEO-MSNs sample used to determine the GC-MS profile of processed PEO? The authors should considered that not only the solvent power of CO2 depends on pressure also selectivity of depends on pressure. In this way, different processed PEO GC-MS fingerprint could be obtained depending of pressure used during the impregnation runs. Thus, the calculation of the impregnation loading should be revised.
  13. – Is PEO chemically interacting with the MSNs or is only physically deposited inside its structure? The authors should incorporate in the manuscript the FTIR analysis of the PEO-MSNs particles and nanocomposite films.
  14. - The authors should compare their results of hygroscopicity and mechanical properties of the films with the reported in other studies.
  15. – The authors should mention in the manuscript which were the conditions of the supercritical impregnation process (cycle index, pressure and time) used to obtained the PEO-MSNs nanoparticles that were used to developed the nanocomposite films.
  16. - Figure 5.a. Is mg of PEO by g of nanocomposite? The author should indicate it in the figure.

Reviewer 2 Report

Page 2, lines 63-64. Rewrite sentence.

Page 2. “How to load EO” change by “EO loading”

Page 2. Emphasize the relation between paragraph in lines 52-66 with the previous work [25].

Page 2, lines 82-84-86. Rewrite sentences.

Caption for Figure 1 is so confusing. Figure should be divided in Figure 1 for a and b, and Figure 2 for c-f.

Some properties are discussed based on the behavior of figures. However, a detailed discussion must be presented. How do the results impact on the final goal of the manuscript? Clarify it. For example page 2 - last paragraph, page 7 – first paragraph,

Page 5, detail a justification for the sentence in lines 142-144.

Explain the term of cycle index.

Figure 3. Clarify units for Cycle.

Page 6. Include in lines 157-159, how about pressure and time?

Figure 5, correct legends.

Page 7-8, lines 187-193. Discussion is about the accumulative released amount in %, however it is expressed as mg/g in Fig 5B, how was it (%) got?

Use homogeneous units, for example mL or ml.

It seems some difficult to understand lines 212 – 217. From Table 2, readers are not able to see cell viability of 84.73% at 20 ug/ml.

I am in doubt about significant digits in Table 2. Report concentration and inhibition rate with the significant digits after deviation, error or uncertainty estimations.

The antibacterial and non-toxic effects should be compared with some finding reported in the literature in order to highlight the present contribution.

Reviewer 3 Report

Overall, the manuscript is of general interest. However, the following issues need to be addressed before the acceptance of the manuscript.

In Fig 1, the Fig C and D should use the same scale bar to compare the MSN before and after the absorption. Meanwhile, for Fig E and F, a top view and a cross-sectional SEM image with a higher resolution is required.

The physical characterization of the film, such as Raman spectra, contact angle, zeta potential and TGA cuvees should be evaluated.

Water flux, NaCl rejection, water permeability, solute permeability, long term separation performance should also be studied.

For the release experiment, the authors should use the percentage of release as the Y-axis.

For the growth inhibition, first, the E coli should also be evaluated. Second, the images with live/dead bacterial should be present.

In all the Figs, the SD should be present where necessary.

Round 2

Reviewer 1 Report

The article is ready for publishing.